# Combined Effects of Clarifying Agents Improve Physicochemical, Microbial and Sensorial Qualities of Fresh Indian Gooseberry (*Phyllanthus emblica* L.) Juice during Refrigerated Storage

**DOI:** 10.3390/foods13020290

**Published:** 2024-01-17

**Authors:** Saeid Jafari, Khursheed Ahmad Shiekh, Dharmendra K. Mishra, Isaya Kijpatanasilp, Kitipong Assatarakul

**Affiliations:** 1Department of Food Technology, Faculty of Science, Chulalongkorn University, Bangkok 10330, Thailand; saeid.j@chula.ac.th (S.J.); khursheedahmad.s@chula.ac.th (K.A.S.); ykijpat@gmail.com (I.K.); 2Department of Food Science, College of Agriculture and Life Sciences, Cornell University, Ithaca, NY 14853, USA; 3Department of Food Science, Purdue University, 745 Agriculture Mall Dr, West Lafayette, IN 47907, USA; mishra67@purdue.edu

**Keywords:** bentonite, characteristics, clarification, gelatin, Indian gooseberry, shelf-life

## Abstract

Using clarifying agents is essential in the production of fruit juice. This study utilized gelatin and bentonite as clarifying agents to improve the quality and shelf-life of Indian gooseberry juice (IGBJ). Different treatments were prepared using varying levels of gelatin and bentonite alone or in combination (1.5–2 mg/mL and 1–2 mg/mL, respectively). The untreated IGBJ was used as a control. The results showed the combined treatment of 1.5 mg/mL gelatin and 1 mg/mL bentonite significantly (*p* ≤ 0.05) improved the transmittance, △E value, total acidity, vitamin C, and antioxidant activity of the IGBJ sample. During storage for 49 days at 4 °C, the quality changes in the IGBJ were minimal with the use of 1.5 mg/mL gelatin and 1 mg/mL bentonite compared to the control (*p* > 0.05). The treated samples showed no signs of spoilage bacteria, yeast, or mold during cold storage. The combined use of gelatin and bentonite (1.5 mg/mL gelatin and 1 mg/mL bentonite) was found to effectively preserve the sensory quality, bioactivity, and color properties of IGBJ, thereby extending its shelf-life. Thus, gelatin and bentonite can be used as preferred filtering aids for quality and shelf-life extension in the food industry, as they have synergistic effects.

## 1. Introduction

In the current scenario of the world, consumer preferences towards healthy food and functional beverages have drastically increased and there was a need to replace chemically preserved food commodities [1]. In response to cater the demand, functional foods and fresh-like beverages are becoming extra popular and widely available in supermarkets [2]. Functional fruit juices contain vitamin C and essential antioxidants that directly strengthen the immune system and directly impose nutraceutical effect on the health and well-being of consumers against dreadful diseases [3]. Apart from their health benefits, their processing and preservation with more innovative approaches are highly recommended worldwide [4]. Several minimal processing techniques such as ultrafiltration, mild thermal, and non-thermal processes including high-pressure processing, ultraviolet irradiation, cold plasma, pulsed electric field, microwave, and ohmic heating have been employed to safeguard the quality of freshly squeezed juices [5]. However, the clarification process is a vital step prior to minimal processing of juices for retaining the bioactive compounds and improving the sensorial quality of fruit juices [6].

The clarification process using fining agents potentially eliminates the turbidity or haze formation due to protein-polyphenol interaction of freshly squeezed raw juice [7]. Gelatin, bentonite, chitosan, silica, and polyvinylpyrrolidone are the most preferred cost-effective clarifying agents in juice processing industry [8]. Gelatin and bentonite absorb suspended particles containing proteins, hydrophobic polyphenols, and cellulose sludge from the raw juice to prevent haze formation after bottling during cold storage. Both clarifying agents pretreated in apple juice were reported to form large aggregates with suspended particles to improve the ultrafiltration process [9]. The positively charged and hydrophobic clarifiers such as gelatin, egg albumin, potassium caseinate, silica gel, and activated carbons were used as the most effective assays in the removal of ochratoxin-A from grape molasses-like syrup [10]. The mechanism of fining agents during clarification process is based on the differences in the nature of ionic charges of protein, polyphenols, and the fining agents, inducing flocculation and sedimentation, resulting in the removal of these potential haze precursors from the juices. Fining or clarification process aids in the removal of active haze precursors thereby decreasing the potential for haze formation during storage, while stabilizing the color and nutritional potential of the juice with appetizing and appealing quality.

Indian gooseberry (IGB) (*Phyllanthus emblica* L.) popularly known as amla contains phenols, antioxidants, and tannins. It is employed in hepatoprotective formulations and has hypoglycemic and hypolipidemic properties [11]. IGB fruits have been reported to have an abundant source of flavonoids (quercetin, kaempferol), phenolic compounds (gallic acid, methyl gallate, ellagic acid, trigallayl glucose), and tannins (emblicanin A and B, phyllaemblicin B, punigluconin, pedunclagin, chebulinic acid, corilagin, geraniin, ellagitannin) [12]. Additionally, these active compounds in the IGB fruit have been reported to treat several non-communicable diseases such as cirrhosis, coronary heart disease, cancer, Alzheimer’s disease, cataracts, arteriosclerosis, and diabetes [13]. However, the richness of vitamin C and bioactive compounds is susceptible to degradation during thermal processing of fruit juices, that affects the quality and shelf-life during storage. Clarification of juice extracted from IGB fruits using fining agents, especially gelatin and bentonite, could be an alternative for the production of haze-free juice product. The clarified juices using fining agents have been referred to as more appetizing during refrigerated storage. 

This is the first report using gelatin and bentonite for haze elimination and quality improvement of IGB juice during refrigerated storage. Therefore, this research aimed to study the effect of clarifying agents (bentonite and gelatin) on the changes in physical, chemical, microbial, sensorial quality, and shelf-life of IGB juice (IGBJ) during storage at 4 °C.

## 2. Materials and Methods

### 2.1. Procurement and Preparation of Juice Sample

The ripe Indian gooseberries (IGB) commonly known as ‘amla’, were obtained in the month of March 2022 from a retail store in Bangkok, Thailand. All the fruit samples were screened based on the uniform size and free of blemish or bruises on the surface. Fruit samples (20 kg) were transported to the Department of Food Technology, Chulalongkorn University. Upon arrival, fruit samples were washed with distilled water to remove dirt and dust, followed by immersion in 0.2 mg/mL of sodium hypochlorite (Merck Co., Darmstadt, Germany) for 2 min, and washed with sterile water prior to peeling and size reduction into slices. The sliced fruit samples were squeezed in a pulp extractor, and the extracted IGBJ was collected in food grade glassware and diluted with sterile water in the ratio of 1:2 (*v*/*v*). The juice (~10 L) was kept at −18 °C until further analysis.

### 2.2. Screening of Fining Agents for Clarification of IGB Juice Samples

The fining process of IGB juice was carried out using clarifying agents such as gelatin (Fujian Funingpu gelatin Co., LTD, Fujian, China) and bentonite (Sigma Aldrich Chemical Co., St. Louis, MO, USA). Gelatin and bentonite clarifying agents were mixed at levels of 1.5 to 2 mg/mL and 1 to 2 mg/mL, respectively. Mixing of clarifying agents was continued with a magnetic stirrer for 30 min at 4 °C. The sludge from the IGBJ samples was centrifuged at speed of 17,387× *g* for 15 min at 4 °C. IGBJ samples after centrifugation were filtered using Whatman^TM^ No.1 filter paper placed on a Buchner funnel attached with a suction vacuum strainer to separate the precipitate. Supernatants of all the IGBJ samples with or without addition of clarifying agents were collected, while the precipitates were discarded. Finally, the IGBJ samples were packed into polyethylene terephthalate (PET) bottles prior to analyses. This study comprised of two experiments. The 1st experiment contained 9 treatments that are detailed as follows: (1) IGBJ-CON: IGBJ without any treatment (control); (2) IGBJ-G1: IGBJ treated with 1.5 mg/mL gelatin; (3) IGBJ-G2: IGBJ treated with 2 mg/mL gelatin; (4) IGBJ-B1: IGBJ treated with 1 mg/mL bentonite; (5) IGBJ-B2: IGBJ treated with 2 mg/mL bentonite; (6) IGBJ-G1+B1: IGBJ treated with 1.5 mg/mL gelatin + 1 mg/mL bentonite; (7) IGBJ-G1+B2: IGBJ treated with 1.5 mg/mL gelatin + 2 mg/mL bentonite; (8) IGBJ-G2+B1: IGBJ treated with 2 mg/mL gelatin + 1 mg/mL bentonite; and (9) IGBJ-G2+B2: IGBJ treated with 2 mg/mL gelatin + 2 mg/mL bentonite. The treated IGBJ samples in the first part of experiment were analyzed for physicochemical and microbial quality immediately after the clarification process. Based on the obtained results from the 1st experiment, the 2nd experiment was conducted with four treatments (i.e., IGBJ-CON, IGBJ-G1, IGBJ-B1, IGBJ-G1+B1 which were evaluated for physicochemical, microbial, and sensorial qualities at 4 °C storage for 49 days. Samples were analyzed every 7 days of storage up to 49 days. However, sensory evaluation was conducted on day 0 and day 49 of refrigerated storage. All the IGBJ treated samples from first and second experiments were filled in sterile PET bottles at a volume of 100 mL, and each sample treatment was prepared in triplicates (n = 3). 

### 2.3. Physicochemical and Microbial Analysis of IGBJ Samples without and with Clarification Process

The pH values and total soluble solid (TSS) content of IGBJ samples treated with gelatin and bentonite were recorded by a pH meter (Eutech Instruments, pH 2700 meter, Ayer Rajah Crescent, Singapore) and a digital refractometer (HI 96801, Hanna Instruments, Nusfalau, Romania), respectively [14]. L* represents lightness from black to white on a scale of 0 to 100, while a* (positive) and a* (negative) indicate red and green values, and b* (positive) and b* (negative) indicate blue and yellow values. Color difference measurement was conducted according to the CIE system (L*, a*, and b*) using Colorimeter (Konica Minolta, modelCR-400, Osaka, Japan) [15]. The color difference was calculated according to the following equation:$$∆E={\left[{\left(L^{*}-L_{0}^{*}\right)}^{2}+{\left(a^{*}-a_{0}^{*}\right)}^{2}+{\left(b^{*}-b_{0}^{*}\right)}^{2}\right]}^{\frac{1}{2}},$$
where ‘L_0_*, a_0_*, b_0_*’ and ‘L*, a*, b*’ presented the color values of IGBJ samples at day 0 and 49 days of refrigerated storage, respectively.

The analysis of vitamin C content was conducted by preparing the standard solution of ascorbic acid (Merck Co., Darmstadt, Germany; 0.1%) and dichlorophenol indophenol (Merck Co., Darmstadt, Germany; DCPIP, 0.1%) [16]. In brief, the ascorbic acid solution was titrated into the DCPIP solution. Then, the volume of ascorbic acid solution required to completely reduce the DCPIP solution was recorded. The titration was repeated multiple times for accuracy, and the average volume of ascorbic acid solution needed to reduce the DCPIP solution was calculated. The vitamin C content was calculated by the following equation:$$\mathrm{Vitamin}\;C\;\mathrm{content}=\frac{\mathrm{Volume}\;0.1\%\;\mathrm{ascorbic}\;\mathrm{acid}\;\mathrm{solution}\;(\mathrm{mL})}{\mathrm{Amount}\;\mathrm{of}\;\mathrm{IGBJ}\;(\mathrm{mL})}\times 100.$$

Total acid (%) was conducted by titration with sodium hydroxide solution as reported by Jafari et al. [17]. Total acid (%) was calculated by the equation as follows:TA (%) = [(C1) × (V1) × (64.04)]/[(V2) × (10)].

TA (%) =total percent acid concentration calculated as an equivalent to citric acid (% *w*/*v*), C1 = sodium hydroxide concentration (N), V1 = sample volume (mL), V2 = volume of sodium hydroxide solution (mL), and 64.04 g = equivalent weight of citric acid.

Total phenolic content, and 2, 2-diphenyl-1-picrylhydrazyl (DPPH; Sigma Aldrich Chemical Co., St. Louis, MO, USA) radical scavenging activity were also conducted as described previously by ourselves [17] and Yıldız et al. [18]. In brief, for DPPH analysis, a 0.5 mL sample was added to 1.5 mL of 1 × 10^−4^ M DPPH. The mixture was allowed to stand in the dark for 15 min, then absorbance was measured at 515 nm using a UV–vis spectrophotometer. An absorbance of DPPH solution and sample was assigned as A_initail_ and A_final_, respectively. A difference was calculated from A_initial_ − A_final_, and used to determine antioxidant activity by DPPH assay. Results were expressed as mol trolox equivalents/L (mol TE/L). Transmittance measurement of IGBJ samples with and without treatments of gelatin and bentonite was performed by using a spectrophotometer (Thermo Scientific GENESYS 20, model: 4001/4, Rochester, NY 14625, USA) at a wavelength of 650 nm.

Total microbial and lactic acid bacteria counts were conducted based on the colony count on the media plates with the reference range of 30–300 colonies. The culture media for determination of total microbial count was prepared by plate count agar (23.5 g; M091-500G, Himedia, Thane, India) dissolved in 1000 mL distilled water and sterilized in an autoclave for 15 min at 121 °C. For total microbial quantification, 1 mL of IGBJ sample was pipetted in a plastic culture dish and pour plate technique was used for enumeration. Then, the dish was incubated for 48 h at 37 °C. For lactic acid bacteria (LAB) counts determination, the culture media was prepared by dissolving MRS agar (67.15 g; M641-500G, Himedia, Thane, India) in 1000 mL distilled water and then sterilized in an autoclave for 15 min at 121 °C. For LAB quantification, 1 mL of IGBJ sample was pipetted in the culture dish with pour plate technique, and the dish was incubated for 48 h at 37 °C. The total microbial and lactic acid bacteria counts were calculated by the equation as follows:CFU/mL = [Number of colonies × dilution factor]/[1 mL].

The mold and yeast counts were analyzed using a spread plate method in which 0.1 mL of IGBJ samples with or without gelatin and bentonite was inoculated. Spoilage yeasts and molds were cultured on a potato dextrose agar (PDA) sterile plastic culture plate [17]. For mold and yeast counts quantification, 0.1 mL of IGBJ sample was spreaded on agar and plastic culture dish was incubated for 48 h at 30 °C. The minimum and maximum limits for yeasts and molds on PDA plate ranged between 30–300 colonies. The yeast and mold colonies were calculated by the equation as mentioned earlier. 

### 2.4. Sensory Evaluation of IGBJ Samples with and without Clarification Process

Sensory evaluation of IGBJ samples was conducted by recruiting 50 experienced panelists with 22–31 years of age (20 males and 30 females) from the Department of Food Technology, Chulalongkorn University. Each panelist received the consent form along with questionnaire prior to samples being served with a three-digit number coded with single-use container. The panelists were asked to express their liking for sample characteristics using a 9-point hedonic scale. IGBJ samples clarified with and without gelatin and bentonite were placed into transparent tasting cups and labelled randomly with three-digit numbers. Sensory scores were given for color, smell, taste, clarity, stability, and overall preferences [19]. The sensory evaluation was approved by the Research Ethics Committee from the office for Human Research Protection in Chulalongkorn University (approval No. 193/2563).

### 2.5. Statistical Analysis

The data were analyzed by analysis of variance (ANOVA) using the computer program Statistical Package for Social Sciences (SPSS Version 23, Armonk, NY, USA), and the mean differences were compared using a Duncan’s multiple range test method at a significance level of *p* ≤ 0.05.

## 3. Results and Discussion

### 3.1. Effect of Clarifying Agents on Physical and Chemical Quality of IGBJ

Table 1 displays the physical and chemical parameter values for the IGBJ-treated samples as well as the untreated control. There were changes in ΔE (1.2–1.7), total acidity (4.2–5.7% citric acid), vitamin C content (22.8–25.4 g ascorbic acid/L), total phenolic content (19.1–19.9 mg GAE/L), and antioxidant activity (130.1–133.4 mol TE/L). The pH ranged from 2.9 to 3.1. The total soluble solid ranged from 19.3 to 20.1 °Brix. Transmittance ranged from 80.3 to 95.8% T at 650 nm. There were no changes (*p* > 0.05) in the pH of the samples. On the other hand, IGBJ-CON sample had higher total soluble solid than treated samples (*p* ≤ 0.05). Total soluble solid content showed no significant differences among the gelatin and bentonite treated IGBJ samples (*p* > 0.05). This could be related to the removal of phenolic glycosides or polysaccharides responsible for turbidity in the juices [9]. Transmittance values were higher in IGBJ-G1+B1 samples (*p* ≤ 0.05) than the IGBJ-CON and treated samples, that confirmed a combination of fining agents potentially eliminated the haziness in the treated IGBJ. Similarly, the color difference values (ΔE) of IGBJ-G1+B1 samples were lower than the IGBJ-CON and other treatments (*p* ≤ 0.05). The IGBJ’s clarity was directly influenced by transmittance and ΔE, meaning that a sample with a lower ΔE value and a higher transmittance had more clarity. It was found the IGBJ-CON samples had significantly lower clarity than all treated IGBJ samples (*p* ≤ 0.05). Gelatin and bentonite at levels of 0.02–0.08 g/L and 1–3 g/L, respectively, were able to retain color properties and increase transmittance of date fruit Kaluteh juice [20]. The clarification and stabilizing properties of gelatin and bentonite facilitate the removal of impurities and particles that might hinder the transmission of light through the juice. This process results in clearer juices with improved color properties and increased transmittance [10]. Bentonite, gelatin, and polyvinylpolypyrrolidone (PVPP) were comparatively studied for the clarification of pomegranate juice, in which bentonite and PVPP fining agents were able to precipitate the impurities from raw juice without altering the total soluble solid, pH and bioactive compounds prior to ultrafiltration process [21].

The IGBJ-G1+B1 contained the highest total acidity and vitamin C contents (*p* ≤ 0.05), compared with the IGBJ-CON and other samples treated with different levels of gelatin and bentonite fining agents. Total phenolic compound was not affected in IGBJ-G1+B1 by the clarification process and showed no differences with that of the IGBJ-CON and other treated samples (*p* ≤ 0.05). However, antioxidant activity was higher in IGBJ-G1+B1 sample than the other samples with or without treatment using gelatin and bentonite fining agents (*p* ≤ 0.05). The improved antioxidant activity could be a result of preserving the original antioxidants present in the fruit due to improved juice clarity [21]. It is important to note the enhancement of antioxidant activity by these agents (i.e., gelatin and bentonite) is often a result of their roles in purification, stabilization, and protection against degradation rather than direct interaction with antioxidants [9]. As an example, gelatin and bentonite can minimize oxidation reactions by removing certain compounds that might promote oxidation or by binding to substances [7]. This helps protect the antioxidants and keeps them from breaking down. The phenomenon governing the fining process of IGBJ might also be related to cationic and anionic charges on gelatin and bentonite, respectively [10]. Moreover, the combination of gelatin and bentonite provided more clarity than the untreated control juice sample because of the aforementioned charge interaction between gelatin and bentonite to precipitate turbid sludge from the raw juice, thereby aiding in the clarification. It has been shown the synergistic effects of gelatin and bentonite help in removing a wider range of impurities, aiding in settling, and improving filtration, contributing to the significantly increased clarity of fruit juice when compared to untreated juice [20]. Gelatin and kaolin were able to precipitate impurities from wine prepared from different fruits such as pineapple, banana, cashew, or pawpaw for effective clarification [22]. Chitosan conjugated with tannic acid was employed as a fining agent to preserve the vitamin C and total phenolics and antioxidant potential of kiwifruit juice [16].

### 3.2. Impact of Clarifying Agents on Physicochemical and Microbial Quality Changes of IGBJ during Storage at 4 °C

Based on the impact of gelatin, bentonite and their combination on the physical, chemical, and sensory attributes of IGBJ, the variants IGBJ-CON, IGBJ-G1, IGBJ-B1, and IGBJ-G1+B1 were chosen for their stability over a 49-day period of cold storage.

The transmittance value of the IGBJ-CON sample was the lowest during all the intervals of storage time compared with the treated samples (Figure 1A; *p* ≤ 0.05). However, the transmittance value of IGBJ-G1+B1 sample was the highest (*p* ≤ 0.05), among all the treated samples and untreated control. IGBJ treated with gelatin and bentonite separately, might have left some protein, polyphenol, and polysaccharide residues creating turbid or hazy appearance of juice during storage. Moreover, the transmittance value corresponds to the clarity of the IGBJ. Therefore, during storage, the clarity of Indian gooseberry or amla juice was reported to decrease, possibly because of protein deterioration or aggregation of proteins, phenolic compounds, and polysaccharide compounds. Chitosan (1.8 g/L) and bentonite (1.0 g/L) were used as fining agents, in which bentonite retained more clarity, as evidenced by higher transmittance in chaenomeles juice and wine clarification during storage [23].

The color difference (∆E) values were highest in the IGBJ-CON sample compared to the treated samples (*p* ≤ 0.05) during the storage period of 49 days, as shown in Figure 1B. The lowest ∆E value, on the other hand, was recorded for the IGBJ-G1+B1 sample (*p* ≤ 0.05). Bentonite and gelatin fining agents in combination worked well to eliminate the suspended particles, including protein residues, polysaccharides, and other compounds responsible for induction of color change in the IGBJ. The ∆E values with bentonite and gelatin treatments were in line with the trend of results obtained in transmittance (Figure 1A), signifying the clarity of juice could be associated with minimum unwanted biochemical residues to trigger a change in color of the IGBJ sample. The color changes and turbidity of clarified banana juice was reduced in bentonite and the combination of gelatin and bentonite fining agents, during six months of storage at 4, 25, or 37 °C [24].

Total soluble solid (TSS) values were higher in control sample without any treatment of fining agents than the treated counterparts (*p* ≤ 0.05), as displayed in Figure 2A. In the case of treated samples such as IGBJ-G1, IGBJ-B1, and IGBJ-G1+B1, no significant difference was analyzed (*p* > 0.05). This could be attributed to the removal of some dissolved solids that might interact with the opposite charged groups of fining agents during the clarification process of IGBJ samples. However, the TSS was highest in the IGBJ+G1 sample among all the treated samples (*p* > 0.05), which might be correlated with the combined effect of gelatin and bentonite to clarify IGBJ sample more effectively than the control sample with higher turbidity. TSS of the apple juice clarified using bentonite, gelatin, and chitosan showed decreased trend during four months of storage due to flocculation and precipitation of dissolved solids at 4 and 20 °C [25].

pH of all samples decreased slowly during storage; however, IGBJ-CON had the highest pH at the end of storage (49 days) as shown in Figure 2C. The steady decrease in the samples might be associated with the protein oxidation and polyphenol hydrolysis under high acidic conditions, generating amine groups that could neutralize the acidic medium [3]. The higher basicity in the samples resisted the lowering of pH values during the entire storage of 49 days at 4 °C. On the contrary, pH of the IGBJ samples treated with gelatin and bentonite or their combination revealed more decreases in pH values, especially in IGBJ-G1+B1 samples during storage compared to control (*p* ≤ 0.05). Low pH levels create an environment is less conducive to microbial growth and proliferation. Microorganisms, including bacteria, fungi, and some viruses, have specific environmental requirements for growth, and pH is a crucial factor influencing their survival and reproduction [3].

Total acidity (TA) of IGBJ-CON and samples treated with fining agents corresponded to the decreased pH values attained in all the samples (Figure 2B). The lowest TA values were observed in IGBJ-CON samples (*p* < 0.05), in comparison with the other treated IGBJ samples. TA of IGBJ samples treated with gelatin and bentonite showed an opposite trend to the pH values. However, the highest TA values were obtained in IGBJ-G1+B1 samples (*p* ≤ 0.05), compared with untreated control. The higher TA values of IGBJ sample using fining agents in combination exhibited high acid groups and the absence of protein amine residues due to efficient clarification process during 49 days of refrigerated storage. The total acidity of juice blend composed of pineapple, carrot and orange juice increased while the pH was decreased during 21 days at 5 ± 1 °C [26].

The vitamin C content of all the IGBJ samples with/without fining process were evaluated during 49 days of refrigerated storage (Figure 2D). Vitamin C content was highest in IGBJ-G1+B1 sample (*p* < 0.05). As shown, the combined effect of fining agents was more pronounced in the retention of vitamin C content in the treated IGBJ sample. The abundance of vitamin C content in IGBJ-CON sample was in line with the TA values (Figure 2B). For the untreated control sample, the presence of oxidative by-products from proteins, polysaccharides, and polyphenols might have degraded the ascorbic acid (AA) content, as evidenced by higher pH and lower TA values. Vitamin C was reported to be stable at low pH medium in kiwifruit juice treated with chitosan-modified tannic acid [16]. Indian gooseberry juice is a rich source of vitamin C responsible for maintaining the quality of juice with high functional value as a ready-to-serve beverage.

Total phenolic compound (TPC) and antioxidant activity of untreated control and IGBJ samples treated with gelatin and bentonite are presented in Figure 3. The highest TPC and antioxidant activity were in the IGBJ-G1+B1 sample compared with the control and other treated samples (*p* ≤ 0.05). During storage, the clarity and stability of the juice play key roles in maintaining the juice’s overall quality, including the retention of antioxidants. The synergistic effects of gelatin and bentonite aid in clarification, which involves removing suspended particles, impurities, and cloudiness from the juice. By doing so, they help stabilize the juice and reduce factors that could lead to degradation or loss of antioxidants over time [20]. TPC in the control sample, which tolerated no clarification process, might be degraded due to oxidative changes during prolonged storage, and the reduction of TPC could correspond to lower antioxidant values. Gelatin and bentonite fining agents showed intermediate TPC and antioxidant activity than the combined IGBJ-G1+B1 sample. The results were in conjunction with the high AA content values obtained during the whole storage of IGBJ-G1+B1 samples as depicted in Figure 2D. The decrease in TPC and antioxidant activity was reported as a result of hydrolysis and the oxidation reaction in pomegranate juice during low-temperature storage [27].

The microbial quality of the clarified IGBJ was conducted on total microbial count, mold or yeast, and lactic acid bacterial (LAB) counts during storage of 49 days at 4 °C (Table 2). All the samples were enumerated for the microbial quality every seven days of storage. Samples clarified with bentonite, gelatin, or their combination displayed no growth of spoilage bacteria, LAB, mold, and yeast until 49 days of refrigerated storage. However, after 49 days of storage, the IGBJ-CON sample displayed 6.8 × 10 CFU/mL of the total spoilage microbial population. Similarly, mold and yeast were visualized in the untreated control sample, 5.2 × 10 CFU/mL at the end of storage. Spoilage LAB was not present in the untreated control and IGBJ samples processed with fining agents. It is crucial to note while gelatin and bentonite might assist in reducing microbial contamination indirectly via particle removal and improved filtration, they are not a substitute for proper sanitation and pasteurization practices necessary to ensure the safety of fruit juice [20]. These results could also be comprehended on the basis of indigenous low pH and lower storage temperature conditions in IGBJ as documented in Figure 2C, for effective microbial growth inhibition. When considering the microbial quality and shelf-life of the IGBJ samples, the untreated control only displayed spoilage microorganisms than the rest of the IGBJ samples clarified with gelatin, bentonite, and their combination. Total bacterial count, including mold and yeast, reported to be inactivated due to high acidified conditions in Indian gooseberry juice [28]. Acidification process and low-temperature storage has been witnessed to inactivate the spoilage bacteria, mold, and yeast in fruit juice [29].

### 3.3. Sensory Evaluation of IGB Juice as Affected by Gelatin and Bentonite Clarification Process during Storage at 4 °C

All the sensory attributes such as color, smell, taste, clarity, stability, and overall preferences were evaluated for IGBJ-CON (without any treatment), IGBJ-G1, IGBJ-B1, and IGBJ-G1+B1 samples (Table 3). The lowest sensory scores were given to the IGBJ-CON sample (*p* ≤ 0.05), especially in taste and clarity parameters compared to IGBJ samples treated with fining agents at day zero of storage (*p* ≤ 0.05). The organoleptic properties of the IGBJ samples treated with fining agents showed similar results irrespective of the clarity visualized in the IGBJ-G1+B1 sample, which was higher than the other treatments (*p* ≤ 0.05).

The IGBJ-CON sample was given the lowest sensory scores as a result of degraded sensory quality at day 49 of storage (*p* ≤ 0.05) compared to the other samples treated with gelatin and bentonite. IGBJ-G1 and IGBJ-B1 samples showed marked differences in all the sensory attributes compared with control sample (*p* ≤ 0.05). Bentonite at 1 mg/mL was able to clarify IGBJ more potentially than gelatin at a treatment level of 1.5 mg/mL. However, gelatin and bentonite levels applied in combination in the IGBJ-G1+B1 sample retained the higher sensorial quality during 49 days of storage at 4 °C. The quality of IGBJ-CON sample was severely degraded due to the oxidative changes in lipids and proteins and other sugar moieties. Even though the gelatin and bentonite acted as promising fining agents, some of the protein or large-sized polyphenols were left as residues in the IGBJ-G1 and IGBJ-B1 samples. In this case, the combination of gelatin and bentonite synergistically clarified the IGBJ more efficiently based on the anionic and cationic charge interactions. Gelatin and bentonite can indirectly contribute to improving the sensorial qualities of fruit juice by enhancing its appearance, texture, and overall consumer experience through clarification and stabilization. In this study, the highest transmittance as well as lowest color difference (ΔE) was recorded for the treated IGBJ samples vs. IGBJ-CON. While gelatin and bentonite themselves might not directly add flavor or aroma to the juice, their role in enhancing clarity, taste, and color indirectly contributes to improving the overall sensorial qualities and consumer acceptance of the fruit juice. Consistently, likeness scores of banana juice clarified with bentonite and gelatin combination presented acceptable scores in terms of color, taste, and overall acceptability during six-month storage at 4 °C [24].

## 4. Conclusions

In fruit juice production, employing clarifying agents is often seen as crucial because clearer juices tend to be more visually appealing to consumers. For this purpose, the optimum level of clarifying agents such as gelatin, bentonite, and their combination were screened on the basis of physicochemical quality of IGBJ. It was noted IGBJ-G1, IGBJ-B1, and IGBJ-G1+B1 samples exhibited the higher light transmittance with minimal changes to the color difference value, TPC, vitamin C, TA, and antioxidant activity, compared with the IGBJ-CON sample. The aforementioned IGBJ samples, including untreated control, were subjected to a 49-day storage study at 4 °C. The IGBJ-G1+B1 sample improved transmittance, TA, vitamin C, TPC, and antioxidant activity compared to other treatments during storage. Consuming fruit juices rich in vitamin C antioxidants can significantly contribute to overall health and wellness by providing a range of these beneficial compounds. The IGBJ-G1+B1 sample maintained a commendable sensory quality on day 49 of storage, achieved through enhancing its clarity, color, and overall visual appeal while minimally affecting its taste or aroma characteristics. As a consequence, gelatin and bentonite could be considered as the preferred filtering aids for clarification resulting in retention of quality and shelf-life extension up to 49 days of storage at 4 °C.

## Figures and Tables

**Figure 1 foods-13-00290-f001:**
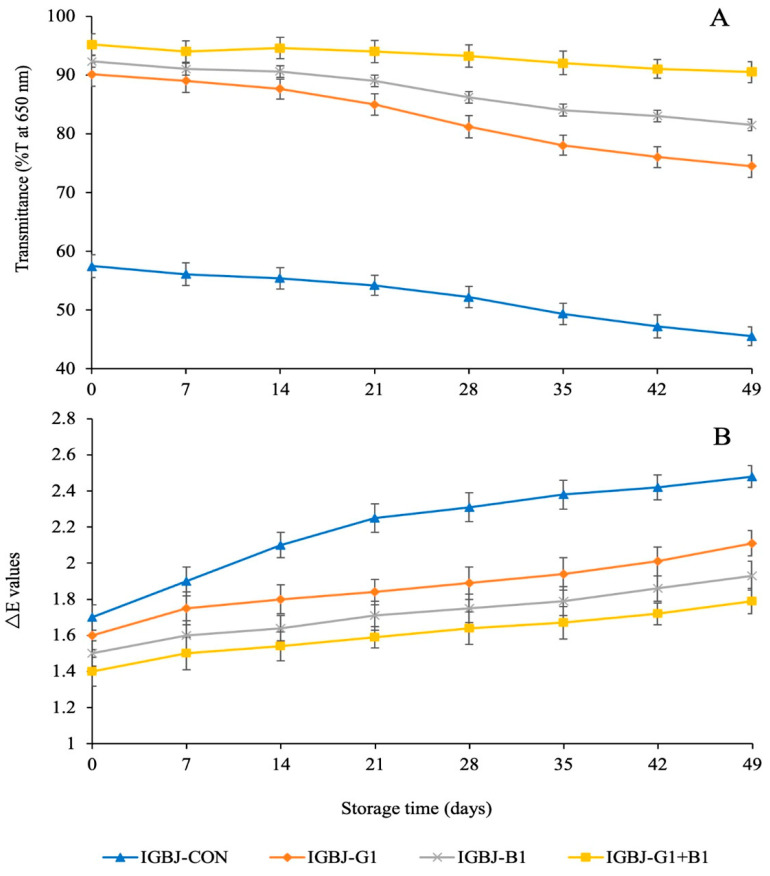
Transmittance (**A**) and ∆E values (**B**) of IGBJ samples during 49 days of storage at 4 °C. Values are presented as mean ± standard deviation (n = 3). IGBJ: Indian gooseberry juice; IGBJ-CON: IGBJ without any treatment; IGBJ-G1: IGBJ treated 1.5 mg/mL gelatin; IGBJ-B1: IGBJ treated 1 mg/mL bentonite; IGBJ-G1+B1: IGBJ treated with 1.5 mg/mL gelatin + 1 mg/mL bentonite.

**Figure 2 foods-13-00290-f002:**
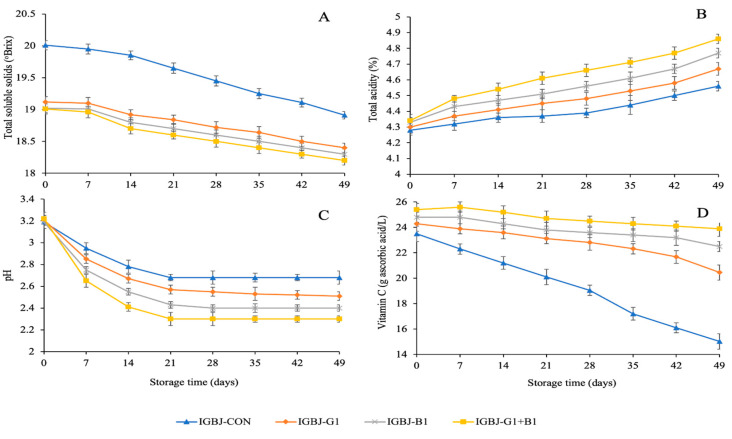
Total soluble solid (**A**), pH (**B**), total acidity (**C**), and vitamin C (**D**) of IGBJ samples during 49 days of storage at 4 °C. Values are presented as mean ± standard deviation (n = 3). See Figure 1 caption for the explanation of treatments.

**Figure 3 foods-13-00290-f003:**
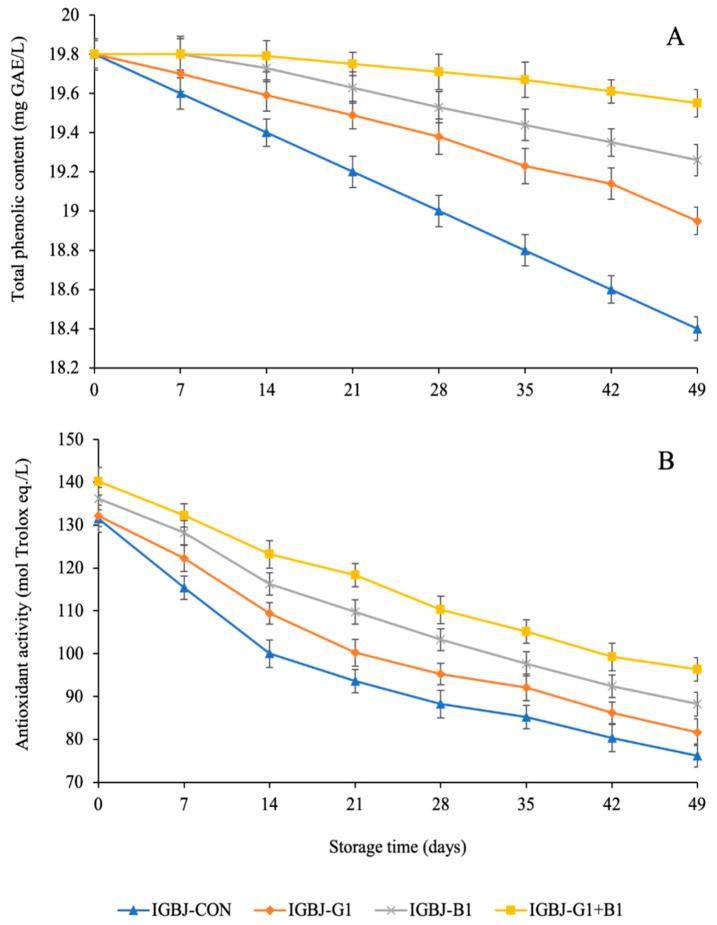
Total phenolic content (**A**), antioxidant activity (**B**) of IGBJ samples during 49 days of storage at 4 °C. Values are presented as mean ± standard deviation (n = 3). See Figure 1 caption for the explanation of treatments.

**Table 1 foods-13-00290-t001:** Effects of gelatin and bentonite clarifying agents on physicochemical properties of Indian gooseberry juice.

Sample Treatments	pH	Total Soluble Solid (°Brix)	Transmittance (%T at 650 nm)	Color Difference Values (ΔE)	Total Acidity (% citric acid)	Vitamin C (g ascorbic acid/L)	Total Phenolic Compound (mg GAE/L)	Antioxidant Activity (mol TE/L)
IGBJ-CON	3.1 ± 0.03 ^a^	20.1 ± 0.1 ^a^	57.6 ± 0.4 ^h^	1.7 ± 0.1 ^a^	4.3 ± 0.03 ^b^	23.5 ± 1.3 ^c^	19.8 ± 0.02 ^a^	131.5 ± 0.03 ^c^
IGBJ-G1	3.1 ± 0.01 ^a^	19.3 ± 0.1 ^b^	90.6 ± 2.7 ^c^	1.6 ± 0.1 ^ab^	5.3 ± 0.03 ^a^	24.3 ± 1.9 ^b^	19.7 ± 0.1 ^a^	131.1 ± 0.02 ^c^
IGBJ-G2	2.9 ± 0.03 ^a^	19.5 ± 0.2 ^b^	81.3 ± 2.5 ^e^	1.7 ± 0.2 ^a^	4.2 ± 0.04 ^b^	22.7 ± 1.6 ^d^	19.1 ± 0.1 ^a^	130.1 ± 0.01 ^d^
IGBJ-B1	3.1 ± 0.02 ^a^	19.4 ± 0.1 ^b^	92.5± 0.4 ^b^	1.5 ± 0.2 ^a b^	5.3 ± 0.06 ^a^	24.8 ± 1.6 ^b^	19.8 ± 0.1 ^a^	132.3± 0.01 ^b^
IGBJ-B2	2.9 ± 0.01 ^a^	19.3 ± 0.1 ^b^	85.7 ± 1.4 ^d^	1.7 ± 0.2 ^a^	4.2 ± 0.07 ^b^	22.8 ± 1.9 ^d^	19.2 ± 0.04 ^a^	130.2 ± 0.01 ^d^
IGBJ-G1+B1	3.2 ± 0.01 ^a^	19.3 ± 0.1 ^b^	95.8 ± 0.9 ^a^	1.2 ± 0.1 ^b^	5.7 ± 0.04 ^a^	25.4 ± 0.9 ^a^	19.9 ± 0.02 ^a^	133.4 ± 0.01 ^a^
IGBJ-G1+B2	2.9 ± 0.01 ^a^	19.4 ± 0.3 ^b^	80.3 ± 5.2 ^g^	1.7 ± 0.2 ^a^	4.2 ± 0.05 ^b^	22.8 ± 1.7 ^c^	19.4 ± 0.03 ^a^	130.4 ± 0.04 ^d^
IGBJ-G2+B1	2.9 ± 0.03 ^a^	19.3 ± 0.1 ^b^	81.1 ± 1.4 ^f^	1.7 ± 0.1 ^a^	4.2 ± 0.03 ^b^	22.8 ± 1.7 ^c^	19.3 ± 0.02 ^a^	130.2 ± 0.01 ^d^
IGBJ-G2+B2	2.9 ± 0.04 ^a^	19.3 ± 0.1 ^b^	80.3 ± 1.2 ^g^	1.7 ± 0.1 ^a^	4.2 ± 0.02 ^b^	22.9 ± 1.9 ^c^	19.5 ± 0.06 ^a^	130.3 ± 0.04 ^d^

Values are presented as mean ± standard deviation (n = 3). Different superscripts of lower-case letters (a–h) within the same column indicate a significant difference (*p* ≤ 0.05). IGBJ: Indian gooseberry juice; IGBJ-CON: IGBJ without any treatment; IGBJ-G1: IGBJ treated 1.5 mg/mL gelatin; IGBJ-G2: IGBJ treated 2 mg/mL gelatin; IGBJ-B1: IGBJ treated 1 mg/mL bentonite; IGBJ-B2: IGBJ treated 2 mg/mL bentonite; IGBJ-G1+B1: IGBJ treated with 1.5 mg/mL gelatin + 1 mg/mL bentonite; IGBJ-G1+B2: IGBJ treated with 1.5 mg/mL gelatin + 2 mg/mL bentonite; IGBJ-G2+B1: IGBJ treated with 2 mg/mL gelatin + 1 mg/mL bentonite; IGBJ-G2+B2: IGBJ treated with 2 mg/mL gelatin + 2 mg/mL bentonite.

**Table 2 foods-13-00290-t002:** Effect of gelatin and bentonite clarifying agents on microbial quality of IGBJ during 49 days of cold storage (4 °C).

		Storage (Days)				
Treatment/Measurement	7	14	21	28	35	49
Microbial load (CFU/mL)						
IGBJ-CON	-	-	-	-	-	6.8 × 10
IGBJ-G1	-	-	-	-	-	-
IGBJ-B1	-	-	-	-	-	-
IGBJ-G1+B1	-	-	-	-	-	-
Mold and yeast (CFU/mL)						
IGBJ-CON	-	-	-	-	-	5.2 × 10
IGBJ-G1	-	-	-	-	-	-
IGBJ-B1	-	-	-	-	-	-
IGBJ-G1+B1	-	-	-	-	-	-
Lactic acid bacteria (CFU/mL)						
IGBJ-CON	-	-	-	-	-	-
IGBJ-G1	-	-	-	-	-	-
IGBJ-B1	-	-	-	-	-	-
IGBJ-G1+B1	-	-	-	-	-	-

- = not detected.

**Table 3 foods-13-00290-t003:** Effect of gelatin and bentonite clarifying agents on sensorial quality of IGBJ during refrigerated storage.

Storage Time (Days)	Treatment	Color	Odor	Taste	Clarity	Stability	Overall Acceptance
0	IGBJ-CON	7.2 ± 0.7 ^Bb^	7.1 ± 0.6 ^Bb^	6.8 ± 0.6 ^Bb^	6.9 ± 0.8 ^Bb^	7.1 ± 0.4 ^Bb^	7.2 ± 0.3 ^Bb^
	IGBJ-G1	8.1 ± 0.4 ^Aa^	8.2 ± 0.3 ^Aa^	8.4 ± 0.4 ^Aa^	8.3 ± 0.4 ^Aa^	8.2 ± 0.2 ^Aa^	8.5 ± 0.2 ^Aa^
	IGBJ-B1	8.4 ± 0.6 ^Aa^	8.3 ± 0.4 ^Aa^	8.5 ± 0.3 ^Aa^	8.4 ± 0.2 ^Aa^	8.6 ± 0.4 ^Aa^	8.6 ± 0.5 ^Aa^
	IGBJ-G1 + B1	8.7 ± 0.3 ^Aa^	8.5 ± 0.2 ^Aa^	8.7 ± 0.5 ^Aa^	8.7 ± 0.3 ^Aa^	8.7 ± 0.3 ^Aa^	8.5 ± 0.2 ^Aa^
49	IGBJ-CON	4.4 ± 0.7 ^Bd^	3.7 ± 0.4 ^Bd^	3.5 ± 0.6 ^Bd^	3.9 ± 0.3 ^Bd^	4.3 ± 0.4 ^Bd^	4.2 ± 0.6 ^Bd^
	IGBJ-G1	5.1 ± 0.3 ^Bc^	5.2 ± 0.3 ^Bc^	5.1 ± 0.4 ^Bc^	5.4 ± 0.5 ^Bc^	5.6 ± 0.3 ^Bc^	5.8 ± 0.5 ^Bc^
	IGBJ-B1	6.2 ± 0.2 ^Bb^	6.4 ± 0.2 ^Bb^	6.3 ± 0.3 ^Bb^	6.7 ± 0.2 ^Bb^	6.5 ± 0.2 ^Bb^	6.4 ± 0.3 ^Bb^
	IGBJ-G1 + B1	7.4 ± 0.5 ^Ba^	7.5 ± 0.4 ^Ba^	7.4 ± 0.5 ^Ba^	7.5 ± 0.3 ^Ba^	7.9 ± 0.5 ^Ba^	7.6 ± 0.4 ^Ba^

Values are presented as mean ± standard deviation (n = 50). Different superscripts of upper (A, B) and lower-case letters (a–d) indicate a significant difference (*p* ≤ 0.05) within the same column and the same row, respectively. IGBJ: Indian gooseberry juice; IGBJ-CON: IGBJ without any treatment; IGBJ-G1: IGBJ treated 1.5 mg/mL gelatin; IGBJ-B1: IGBJ treated 1 mg/mL bentonite; IGBJ-G1+B1: IGBJ treated with 1.5 mg/mL gelatin + 1 mg/mL bentonite.

## Data Availability

Data is contained within the article.

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
