# Peer review of "Combined Effects of Clarifying Agents Improve Physicochemical, Microbial and Sensorial Qualities of Fresh Indian Gooseberry (Phyllanthus emblica L.) Juice during Refrigerated Storage"

_foods, 2024, doi:10.3390/foods13020290_

Round 1
Reviewer 1 Report
Comments and Suggestions for Authors
Comments and suggestions for authors are in the pdf file

Minor editing of English language required
Author Response
|
Introduction: • There are problems with misspellings and joined words.
• It cannot be a recommendation not to use harmful additives; it is a necessity to protect consumers.
• Lines 62-65: Excessive use of capital letters |
· All the misspellings and joined words have been revised.
· The sentence about “harmful additives”has been revised for
· Revised. |
|
Materials and methods: • Write amla and not 'Amla', it is not a cultivar but the local name of this fruit.
• The heading Materials and methods should be numbered with 2 and not 2.0. • Section 2.2 is missing.
• As this juice is not very well known, I consider it necessary to provide the color data (L*, a* and b*), at least for the product on day 0, just obtained, with and without clarification treatments.
• The design of the experiment should be clarified,
• How many replicates and of what volume were the different treatments?
• I cannot know at what stage of ripening the fruit is at and how much fruit weight is taken, and how much juice is made for all analyses, including sensory analysis.
• I think I understand that two experiments were carried out, the first one characterizes the different juices, with two different concentrations of two clarifiers and their combinations (9 treatments) and the second one, with a smaller number of samples (4 treatments), carries out the shelf-life study and the sensory evaluation. If this is the case, it should be clarified.
• The information from Lines 102 to 107 would be more organized and easier to read if it were put in a table, or with another way of designating the treatments.
• I recommend improving the explanation of the methodology for the analysis of vitamin C and microorganisms.
• I also recommend using the Pearson correlation to establish relationships between variables on an objective basis. |
“amla” has been inserted.
· “2” has been inserted. · The sections are re-numbered.
· We preferred to use “ΔE” in this study to show the color differences. The reason for this was that “ ∆E value between 1.0 and 2.0 suggests a minimally detectable color change, while a value between 4.0 and 5.0 indicates a significant color difference. Since the treatments made non-significant changes and between 1.0 and 2.0, so we refrained from including L*, a* and b*.
· Details have been clarified in the experimental design. Please see lines 100-116. Thanks.
· Each treatment had 3 replicates and each replicate was of 100 mL volume packed in PET bottles.
· We used “ripe fruit” in this study as shown in section of “Procurement and preparation of juice sample”. In this study, we used almost 20 kg of fresh amla fruit and we extracted about 500 milliliters of juices/kg of fruit.
· A more explanation has been inserted in the text for more information.
· The lines “102-107” have been revised such as numbering the treatments as well as more explanations on the procedure for better understanding.
· More explanations of the methodology for the analysis of vitamin C and microorganisms have been inserted.
· We appreciate your recommendation. However, the Pearson correlation is not in the scope of the present study. We will consider it for our future studies. |
|
Results and discussion: • 173-178 rewrite more clearly.
• 181, 270 and others: significant, not marked
• 182 Add a bibliographic reference to supporting this hypothesis
• 185 Delta
• 187 Corrected with???? (Pearson correlations)
• 204-206 The increment in the antioxidant could be associated with the exposure of sufficient functional groups in the polyphenols due to removal of proteins and other tannins responsible for protein-polyphenol interactions. The sentence is not clear and is either a hypothesis supported by literature or a conclusion of the authors.
• 250-252 Clarify sentence
• Fig. 1B Put the same format on the axis as in Fig. 1A.
• Title of Fig.1 treated without???? gelatin and bentonite
• Fig 2 has poor resolution, difficult to read it
• 282-283 Bibliographic citation required
• 298-302 Clarify paragraph
• 313 Should read Fig 1
• 317-320 It is not clear
• Along the text, the authors use shelf-life and shelf-life, please, unify
• 380 the least sensory properties is not an appropriate way to write about results.
• Table 3. Arrange the table footer and the lines |
“173-178” have been rewritten.
“Significant” has been replaced.
“A bibliographic reference” has been added.
“ΔE” has been inserted.
The line 187 has been deleted to prevent from misunderstanding.
The line 204-206 has been revised for more understanding.
The line “250-252” has been revised.
The same axis format has been given for Fig.1A and Fig. 1B.
The title of Fig.1 has been improved.
Fig 2 has been improved.
A reference has been inserted for line 282-283.
Line 298-302 has been revised for more understanding.
It has been corrected.
Line 317-320 has been revised for more understanding.
“shelf-life” has been inserted in whole document.
Sentence has been corrected for intended meaning. Line 382-383.Thanks.
The Table 3 footer has been rearranged. |
|
Conclusions: • should be more concrete and written with more quality • 435 cost? • 432-433 Do you mean “compared to the rest of the treatments”, please clarify it. |
The conclusion section has been improved by considering the reviewer’s comments. |

Reviewer 2 Report
Comments and Suggestions for Authors
The authors utilized gelatin and bentonite as clarifying agents to improve the quality and shelf-life of Indian gooseberry juice (IGBJ). The authors accomplished the study by preparing different treatments and varying levels of gelatin and bentonite alone or in combination (1.5-2 mg/mL and 1-2 mg/mL, respectively).Results showed were very promising, and appear well discussed. Just a few areas seem unclear, as below:
a)In 3.1., and 3.2 sections, why are clarifying agents impact on quality of IGBJ so distinct? I could not see it in these discussions. Please, expand the discussions more on the 'why', not only on what A or B said
b) In 3.3, the sensorial discourse seems scanty. There is no linkage with other tested quality attributes. Please make some connections ok
Look forward to your revised manuscript
Author Response
|
Comments from the 2ndreviewer |
Corrections made |
|
The authors utilized gelatin and bentonite as clarifying agents to improve the quality and shelf-life of Indian gooseberry juice (IGBJ). The authors accomplished the study by preparing different treatments and varying levels of gelatin and bentonite alone or in combination (1.5-2 mg/mL and 1-2 mg/mL, respectively). Results showed were very promising, and appear well discussed. Just a few areas seem unclear, as below: |
We would like to thank the reviewer for his/her positive views on our article. |
|
a) In 3.1., and 3.2 sections, why are clarifying agents impact on quality of IGBJ so distinct? I could not see it in these discussions. Please, expand the discussions more on the 'why', not only on what A or B said b) In 3.3, the sensorial discourse seems scanty. There is no linkage with other tested quality attributes. Please make some connections ok |
a) The sections 3.1 and 3.2 have been improved according to the reviewer’s comments.
b) More explanations on the connection of sensorial qualities and other properties like color difference (ΔE) and the transmittance have been given. |
